# Expanding Horizons of CRISPR/Cas Technology: Clinical Advancements, Therapeutic Applications, and Challenges in Gene Therapy

**DOI:** 10.3390/ijms252413321

**Published:** 2024-12-12

**Authors:** Ahmad Bairqdar, Polina E. Karitskaya, Grigory A. Stepanov

**Affiliations:** 1Institute of Chemical Biology and Fundamental Medicine of the Siberian Branch of the Russian Academy of Sciences, Novosibirsk 630090, Russia; ahmad15.berkdar@gmail.com; 2Department of Natural Sciences, Novosibirsk State University, Novosibirsk 630090, Russia; p.karitskaya@g.nsu.ru

**Keywords:** CRISPR–Cas, gene therapy, xenotransplantation, genome editing

## Abstract

CRISPR–Cas technology has transformed the field of gene editing, opening new possibilities for treatment of various genetic disorders. Recent years have seen a surge in clinical trials using CRISPR–Cas-based therapies. This review examines the current landscape of CRISPR–Cas implementation in clinical trials, with data from key registries, including the Australian New Zealand Clinical Trials Registry, the Chinese Clinical Trial Register, and ClinicalTrials.gov. Emphasis is placed on the mechanism of action of tested therapies, the delivery method, and the most recent findings of each clinical trial.

## 1. Introduction

The discovery and development of genome editing systems have expanded the possibilities for managing and precisely “tuning” eukaryotic genome sequences [1,2,3]. DNA editing based on programmable nucleases in human gene therapy has led to several advancements, allowing for more precise correction of genetic mutations, the removal of “undesirable” genome elements, or the addition of new sequences to achieve therapeutic effects [4,5].

Many genome editing methods rely on the introduction of a precise double-strand break (DSB). Currently, four editing systems have been identified for inducing site-specific DSBs: meganucleases (homing nucleases), zinc-finger nucleases (ZFNs), transcription activator-like effector nucleases (TALENs), and CRISPR–Cas systems [6]. After introducing a double-strand break via any of these systems, DNA integrity is restored through one of two main pathways: homologous recombination repair (HDR) or non-homologous end joining (NHEJ) [7].

Genome editing systems like meganucleases, ZFNs, and TALENs require researchers to design new protein domains for each target. In contrast, the CRISPR–Cas system is one of the most accessible and easiest to use, as it only requires the design and construction of a guide RNA (sgRNA) for specific targeting [8].

This review examines the application of CRISPR–Cas systems in clinical trials for treating genetic disorders, cancers, and infectious diseases. By analyzing data from clinicaltrials.gov and ChiCTR.org, it explores targeted genes, trial outcomes, and key challenges, offering insights into the current and future potential of CRISPR-based therapies.

## 2. CRISPR–Cas as a Prokaryotic Adaptive Immune System

Bacteriophages and phages, which are widespread on our planet, pose a constant threat to the life of prokaryotes, prompting them to enhance their defense mechanisms. One of the most interesting defense strategies is the CRISPR–Cas system, which was first described in 1987 [1].

Prokaryotic CRISPR loci consist of clustered short palindromic repeats interspersed with variable DNA sequences (spacers) [9]. Various Cas genes are located adjacent to the CRISPR locus, together forming the CRISPR–Cas system. Present in many archaea and bacteria, CRISPR–Cas loci function as adaptive immune systems, targeting and breaking down foreign genetic material that enters the cell [10].

Fragments of foreign DNA (protospacers) are integrated into the CRISPR locus and become new spacers. These spacers provide immune memory for the bacterium or archaea, protecting against future invasions by similar genetic material [11]. Pre-crRNA is transcribed from the CRISPR locus and then processed into crRNA, which forms a complex with Cas proteins, guiding them to specifically cleave targeted DNA or RNA [12].

## 3. Modern Classification of CRISPR–Cas Systems

As of 2020, the number of newly characterized CRISPR–Cas systems continues to grow, thanks to advances in bioinformatics approaches and increased interest in this system, particularly as an efficient method for genetic modification for various purposes [13].

According to the 2020 classification, CRISPR–Cas systems are divided into two classes, which include six types and 33 subtypes. The basis for classifying systems into classes and types is the structure of the effector complexes (proteins involved in cleaving the target). Class 1 CRISPR–Cas systems contain multiple Cas proteins that form effector modules, while Class 2 systems consist of a single multidomain effector crRNA-binding protein that is functionally analogous to the multiple effector modules of Class 1 [14].

Despite their prevalence in nature, Class 1 CRISPR systems face challenges in clinical use due to their multi-protein nature, which complicates delivery into cells [15]. On the other hand, CRISPR–Cas Class 2 systems are simpler and more widely used in gene editing technologies, and they are more favored for clinical use due to their simpler architecture and ease of manipulation compared to Class 1 systems [16]. Since Class 2 CRISPR systems consist of a single, multidomain effector protein that can target DNA or RNA, they are easier to deliver into cells. This class is further divided into several types [14]:Type II Systems (Cas9): The most well-known CRISPR system. Cas9 recognizes and cleaves target DNA with high specificity, guided by a small guide RNA (gRNA) [17]. This system has been widely adopted for gene therapy, diagnostics, and research, as described in the following section.Type V Systems (Cas12): Cas12 is similar to Cas9 but has unique properties, such as its distinct ability to create single-stranded overhangs when cutting DNA. This feature enhances its precision for gene editing applications, allowing for more controlled deletions, insertions, and targeted modifications [18].Type VI Systems (Cas13): CRISPR–Cas13 systems are the newest and best characterized single-effector nucleases of CRISPR (Figure 1), whose target is single-stranded RNA. The Cas13 enzyme is an RNA-guided RNA-specific nuclease [19]. CRISPR–Cas13d systems are a promising tool for knocking down specific RNA sequences without altering the genome, offering a safer alternative in situations where permanent DNA changes are undesirable [20].

## 4. Clinical Implementations

### 4.1. Cancer Therapy

Cancer therapy is currently the most prominent field where CRISPR–Cas technology is being applied [31]. The current use of the CRISPR–Cas system in cancer therapy can be summarized in three main aspects: oncogene knockouts, correcting tumor suppressor genes, and modifying immune cells used in adoptive immune therapy [32]. However, only the third strategy is currently used in clinical trials and has been shown to be an effective approach in the genome editing of immune cells, particularly Chimeric Antigen Receptor T (CAR-T) cells [33].

The modification of immune cells applied in adoptive therapies aims to achieve three main goals. First, making those cells less likely to be rejected by the recipient and reducing the severity of the graft-versus-host reaction. Secondly, reducing the probability of these cells being deactivated by highly mutated tumor cells, thereby lowering the tumors’ ability to evade detection [34]. Finally, reducing fratricide, especially in hematological malignancies [35].

NCT04037566 is one of the earliest trials implementing CRISPR–Cas9 to edit hematopoietic progenitor kinase 1 (*HPK1*) in CD19 CAR-T cells in patients with relapsed/refractory (R/R) leukemia or lymphoma [36]. Another very popular target of Cas9-mediated knockouts in T cell-based adoptive immunotherapy is programmed cell death protein 1 (*PD-1*) and the endogenous T cell receptor gene (*TCR*) [37].

By reviewing and analyzing data from (ClinicalTrials.gov), we found that the genes targeted and knocked out by CRISPR–Cas systems are similar across many trials. Table 1 summarizes the implemented CRISPR–Cas systems and their gene targets for various registered clinical trials directed for different types of malignancies.

Most recently, CRISPR–Cas13 has been investigated as an alternative for Cas9-based genome modifications of T cell-based adoptive immunotherapies, where Cas13 was used for disrupting the expressed transcript of key immunoregulatory genes resulting in enhanced T cell fitness and anti-tumor activity both in vitro and in vivo [38].

**Table 1 ijms-25-13321-t001:** Clinical trials employing CRISPR–Cas for cell editing in adoptive immunotherapy of different types of malignancies.

Clinical Trial ID	Cancer Type	CRISPR–Cas System/Target	Modified Cells	Reference
NCT04037566	B and T cell malignancies/renal cell carcinoma	CRISPR–Cas9 disrupting TCR and *MCH I* ^1^	Allogeneic CAR-T cells	[37]
NCT04035434	Non-Hodgkin and large B cell lymphomas	CRISPR–Cas9 disrupting TCR and *MCH I*	Allogeneic CAR-T cells	[39]
NCT05942599	Acute myeloid leukemia	CRISPR–Cas9 disrupting CD33	Allogeneic CAR-T cells	[40]
NCT05397184	Acute myeloid leukemia	CRISPR–Cas9 disrupting CD7, CD52, and TCR	Allogeneic CAR-T cells	[41]
NCT04426669	Gastro-intestinal Cancer	CRISPR–Cas9 *CISH* ^2^ inactivation	Tumor-infiltrating lymphocytes	Clinicaltrials.gov
NCT05566223	Non-small cell lung cancer	CRISPR–Cas9 CISH inactivation	Tumor-infiltrating lymphocytes	Clinicaltrials.gov
NCT05795595	Relapsed or refractory solid tumors	CRISPR–Cas9 CRISPR–Cas9 disrupting TCR and MCH I and CD70	Allogeneic CAR-T cells	[42]

^1^—Major histocompatibility complex-I, ^2^—Cytokine inducible SH2-containing protein.

All in all, the clinical application of the CRISPR–Cas system in the field of oncology is still almost exclusively limited to adoptive immunotherapy. The source of the infused modified cells is usually allogeneic, especially in cases of blood cancers where, in most cases, immune cells are the target of the therapy.

### 4.2. Blood Disorders

Beta-thalassemia and sickle cell anemia are both Mendelian single-gene disorders, affecting the adult β hemoglobin gene (*HBB*) [43]. For beta-thalassemia, there are over 300 known alleles causing the condition, with mutations ranging from single nucleotide polymorphisms to small deletions, and in rare cases, wide deletions or insertions of transposable elements [44]. These mutations result in different phenotypes, from β0, where no expression of the beta chain occurs, to β+, where beta chain expression is at insufficient levels, and more rarely, β++, where only a mild deficiency of the beta chain is seen [45]. On the other hand, sickle cell anemia is associated with a single point mutation (β codon 6, GAG to GTG), which leads to the expression of hemoglobin S and the complex symptoms related to its expression [46].

The field of hematological disease witnessed the first CRISPR-based genetic therapy approved by the Food and Drug Administration (FDA). The drug, CTX001 (now marketed as Exa-cel), was developed collaboratively by Vertex and CRISPR Therapeutics [47]. Several clinical trials were already underway prior to the approval of CTX001, at the end of 2023. These trials aim to enhance treatment efficiency and investigate alternative approaches that avoid the patented design of the original drug.

All current attempts (Table 2) to develop genetic therapies for beta hemoglobin-related diseases rely on three main approaches:Activation of the fetal gamma hemoglobin gene (*HBG1/2*) within the beta-globin cluster by silencing the B cell lymphoma/leukemia 11A gene (*BCL11A*), a transcription factor that acts as a repressor when adult hemoglobin is expressed [48].Modifying a specific site in the promoter of the gamma hemoglobin gene prevents BCL11A from binding, leading to the activation of the gamma hemoglobin gene [49].Homologous recombination-based repair of the *HBB* gene by CRISPR–Cas9-induced double-strand breaks followed by introducing a DNA template to guide the repair process. [50].

Therapies using the first approach typically employ Cas9 to target and silence the *BCL11A* gene by inducing a cut at the GATA1/TAL1 binding sites of the *BCL11A* erythroid specific enhancer (Figure 2). Examples include CTX001, BRL101, and ET01, currently in clinical trials (NCT06287099, NCT04925206). These therapies use electroporation to deliver Cas9 and its guide RNA in vitro, followed by the reinfusion of the modified and sorted cells into the patient [51,52,53].

Other trials seek to activate gamma hemoglobin expression by mutating the promoter region of the HBG1/2 gene, altering the BCL11A interaction site to activate the gene. These trials are still in clinical phases (NCT05444894, ChiCTR2100053406, NCT06041620), with drugs such as EDIT-301, RM-001, and VGB-Ex01. RM-001 uses Cas9, while the other drugs use Cas12 to induce cuts in the promoter zone [54,55]. Despite some differences between Cas9- and Cas12-like types of induced cut and targeted sequences [56], the main goal in this case is to develop a new editing tool using different Cas proteins to evade existing patents.

Finally, GPH101, developed by Vertex, employs Cas9-induced breaks to target the mutant allele of the *HBB* gene around the mutation site and introduces a template for homology repair. GPH101 is still in early clinical phases (NCT04774536), and this approach is still limited to cases of single nucleotide changes leading to the expression of hemoglobin S [57].

**Table 2 ijms-25-13321-t002:** Ongoing clinical trials developing CRISPR–Cas-based gene editing therapies for beta-thalassemia and sickle cell anemia.

Clinical Trial ID	Blood Disorder	CRISPR–Cas System/Target	Treatment ID	Reference
NCT05477563 NCT05356195	Beta-Thalassemia Sickle Cell Disease	CRISPR–Cas9 disruption of BCL11A	CTX001	[53]
ChiCTR2100053406	Beta-Thalassemia	CRISPR–Cas9 disruption of HBG1/HBG2 promoters	RM-001	[55]
NCT06041620	Beta-Thalassemia	CRISPR–Cas12b disruption of HBG1/HBG2 promoters	VGB-Ex01	Clinicaltrials.gov
NCT04819841	Sickle Cell Disease	CRISPR–Cas9 correction of HBB	GPH101	[57]
NCT06314529 NCT06300723	Beta-Thalassemia Sickle Cell Disease	CRISPR–Cas9 disruption of BCL11A	BRL-101	[52]
NCT05444894 NCT06363760	Beta-Thalassemia Sickle Cell Disease	CRISPR–Cas12b disruption of HBG1/HBG2 promoters	EDIT-301	[54]
NCT04925206	Beta-Thalassemia	CRISPR–Cas9 disruption of BCL11A	ET-01	[51]

The use of the CRISPR–Cas system in hematological disorders has eliminated the need for recurrent blood transfusions and reduced the risk of associated coagulation and blood clot formation. Additionally, the success of autologous cell modification by the CRISPR–Cas system in beta-thalassemia and sickle cell anemia has revitalized the field of autologous stem cell transplantation, which are known to have lower risks of rejection, in various fields of regenerative therapy [58,59].

### 4.3. Inflammatory and Metabolic Diseases

The contribution of genetic factors to inflammatory and metabolic disorders varies significantly among individuals and is often disease-specific. However, certain mutations in a single gene can be directly linked to well-characterized disorders. CRISPR–Cas-based therapy is being investigated for specific inflammatory and metabolic conditions. CTX320 (ACTRN12623001095651p), a CRISPR–Cas9-based therapy, is designed for the treatment of atherosclerosis and other conditions related to high levels of lipoprotein (a) (Lpa) [60]. The drug knocks out the apolipoprotein (a) gene (*APOA1*) specifically in the liver, using lipid nanoparticles for targeted delivery. Knocking out *APOA1* leads to the disruption of the apolipoprotein (a) component of lipoprotein (a), resulting in its reduced formation and lower Lpa blood levels [61].

The same company is developing another drug, CTX310 (ACTRN12623000809639), a gene editing therapy aimed at treating refractory dyslipidemia by targeting angiopoietin-like protein 3 (*ANGPTL3*). ANGPTL3 is synthesized exclusively in the liver and is a critical regulator of lipoprotein metabolism [62]. Studies in animals and humans have shown that loss of function or downregulation of *ANGPTL3* is associated with an improved lipid profile and a lower risk of atherosclerotic lesions and cardiovascular events [63]. CTX310 uses targeted delivery of the CRISPR–Cas system via lipid nanoparticles to knock out *ANGPTL3* in the liver [64].

Another study, still in preclinical phases, is testing a Cas9-based drug targeting matrix metalloproteinase-13 (MMP-13) (NCT03057912) expression in chondrocytes. MMP-13 is the primary matrix metalloproteinase involved in cartilage degradation due to its ability to cleave type II collagen, making it an attractive target for the treatment of osteoarthritis [65]. The delivery method involves intra-articular infusion of Cas9, encapsulated in modified liposomes that carry a chondrocyte affinity protein (CAP) to specifically target MMP-13-expressing chondrocytes [66].

Cas9 is also being actively investigated for therapy in type 1 diabetes (T1D), especially after the relative success of stem cell therapy for T1D by Vertex [67]. To improve the efficiency of differentiated beta cells, Vertex partnered with CRISPR Therapeutics to modulate stem cells used in therapy to evade immune responses (NCT05565248, NCT05210530). The targeted genes are beta-2-microglobulin (*B2M*) and thioredoxin interacting protein (*TXNIP*) [68]. The general approach used for the modification of allogenic stem cells in similar studies is (1) knockout of immune-stimulating genes using CRISPR–Cas nuclease activity or (2) knock-in of immune-evasiveness genes using deadCas9 (dCas9) protein for targeted delivery [69].

### 4.4. Infectious Diseases

Therapeutic strategies for infectious diseases implementing CRISPR systems currently explore two approaches: (1) cleaving drug-resistant genes in bacteria, and (2) cleaving viral genomes integrated into infected cells [70]. However, in clinical trials only the second approach is being investigated.

#### 4.4.1. Human Immunodeficiency Virus (HIV)

The C-C chemokine receptor type 5 (CCR5) plays a key role in HIV infection due to its involvement in the viral entry process [71]. The discovery of naturally occurring CCR5 mutations has allowed scientists to target CCR5 as a potential way to prevent or limit HIV infection in vivo [72]. CRISPR–Cas paved the way for the long-awaited HIV cure, with promising results from a Chinese clinical study involving an HIV patient with lymphoblastic leukemia (NCT03164135). The patient received a transfusion of hematopoietic stem cells with CCR5 gene ablation, resulting in remission. The CCR5-ablated CD4+ cells persisted for 19 months without any gene editing-related adverse effects. However, the knockout efficiency of the transfected cells was around 5%, and the ex vivo modulation and reinfusion process complicated the treatment [73].

The Chinese group’s approach focused on developing HIV-resistant CD4+ cells, which was proved to be a challenging. In contrast, Excision BioTherapeutics adopted a more conventional strategy with EBT-101, a Cas9-based therapy designed to target and cleave proviral DNA. This therapy is delivered via an in vivo infusion of adeno-associated virus (AAV9) encoding the Cas9 protein. Preclinical studies in non-human primates showed promising results [74], and a recent press release announced that EBT-101 had no adverse effects and only mild side effects in the ongoing (NCT05144386) phase 1/2 clinical trial [75].

#### 4.4.2. Human Papilloma Virus (HPV)

HPV is the most common viral infection of the reproductive tract [76], with strong associations to cervical cancers in females [77]. Current prophylactic vaccines against HPV primarily target HPV types 16 and 18 [78]. A new genetic therapy based on TALEN and Cas9 is being developed for the treatment of HPV16 and HPV18 (NCT03057912). This therapy utilizes the nuclease activity of both TALEN and Cas9 to disrupt the E6 and E7 DNA integrated into the squamous cells of the cervix by in vivo plasmid delivery. Despite the positive in vitro results obtained by the group [79], the findings of the clinical trial were not published.

#### 4.4.3. Herpes Simplex Virus 1 (HSV-1)

HSV-1, another virus from the herpesvirus family, can lead to keratitis, a common cause of corneal ulcers and blindness [80]. Despite available treatments, this type of keratitis can recur and become refractory to medications [81]. Cas9 is currently being investigated as a potential therapeutic option for HSV-1 complications (NCT04560790). A Chinese group studied the injection of lentiviral particles expressing CRISPR–Cas9 and guide RNA targeting the viral DNA UL8 and UL29 into the stromal part of the cornea of a transplant patient. The treatment showed no adverse effects, and no off-target DNA cleavage was detected. All three patients showed no traces of viral DNA in tear swabs after 18 months of Cas9 treatment, even without antiviral medications [82].

### 4.5. Others

CRISPR–Cas can be the therapy of choice for tackling various rare genetic diseases due to its simplicity and programmability [83]. Multiple clinical trials have been initiated to test the safety of CRISPR-based gene therapy in patients with rare genetic disorders, with some drugs showing promising results and accelerating through development stages (Table 3).

Hereditary angioedema (HAE) is a rare genetic disorder, with a frequency of 1 in 50,000, that causes sudden, severe episodes of angioedema [84]. HAE is classified into two main types: HAE due to C1-esterase inhibitor (C1INH) deficiency and HAE with normal C1INH levels. The first type is characterized by an autosomal mutation in the serpin family G member 1 (*SERPING1*) gene, which encodes C1-INH, leading to uncontrolled activation of plasma kallikrein, which, in turn, causes excessive production of bradykinin. [85]. Intellia Therapeutics has already published results from their phase 1 clinical trial (NCT05120830) testing the safety and tolerability of NTLA-2002, a CRISPR–Cas9 in vivo editing drug targeting and silencing the kallikrein B1 (*KLKB1*) gene, responsible for bradykinin breakdown [86]. Results showed no safety concerns, with the reduction of kallikrein levels correlating with the dose, and patients experiencing a 95% reduction in angioedema attacks during the month of observation [87].

NTLA-2001 is another drug developed earlier by the same company for treating hATTR amyloidosis (NCT04601051). The drug uses the same gene editing and delivery method, with the only difference being the guide RNA sequence. This sequence targets the transthyretin (*TTR*) gene, which produces the TTR protein, whose misfolding and accumulation lead to multisystem dysfunction [88]. The drug induces a break in the TTR gene, leading to a 52% reduction in TTR expression at low doses and an 87% reduction at higher doses [89].

Another clinical trial, started in 2023, is testing ZVS203e, a Cas9-based gene therapy for retinitis pigmentosa. It targets the rhodopsin (*RHO*) gene with site-specific mutation and is delivered by subretinal injection of an AAV vector. The drug is still in phase 1, with no results yet published (NCT05805007).

In May 2024, a significant development opened the door for a promising treatment for CEP290-related retinal degeneration. EDITAS Medicine initiated a clinical trial involving 15 patients over 3 years. Patients were divided into groups receiving different doses of EDIT-101, a Cas9-based gene editing therapy delivered via subretinal injection of AAV expressing the CRISPR–Cas9 system targeting the mutated allele [90]. Previous results showed that deleting the mutated region of the IVS26 allele of the *CEP290* gene through CRISPR–Cas9-mediated non-homologous end joining (NHEJ) restores normal splicing of the transcript and the expression of the normal CEP290 protein. No serious adverse events or dose-limiting toxic effects were recorded, and preliminary results indicated significant improvements in vision and quality of life for most participants [91].

### 4.6. CRISPR–Cas13 in Clinical Trials

As mentioned above, the Cas13 is the best characterized Cas protein, whose target is single-stranded RNA. In addition to its molecular advantage of being able to process ribonucleic acids [29], Cas13 introduces another important feature related to its potential clinical applications. Since Cas13 modifies RNA transcripts, its effects on the human genome can be avoided [92], and permanent alterations in the DNA sequence can be replaced by either permanent or transient knockdown of gene expression.

Most studies on Cas13 are still in the preclinical trial phase, either in vitro or in vivo. The only registered clinical studies on clinicaltrials.gov are currently being conducted in China by HuidaGene (NCT06031727, NCT06025032). The first trial aims to treat age-related macular degeneration (AMD), a common multifactorial eye disease affecting the central area of the ocular posterior segment [93]. This condition is characterized by the growth of atypical choroidal vessels due to increased expression of vascular endothelial growth factors (*VEGF*) [94]. HG202, a treatment for neovascular AMD still under development, uses high-fidelity CRISPR–Cas13 to knock down VEGF expression. The Cas13 protein is delivered via an adeno-associated viral vector by subretinal injection [95].

The second Cas13-based therapy, HG205, aims to treat congenital hearing loss through intracochlear injection of AAV expressing Cas13. This drug is already in phase 1 clinical trials. However, primary results have yet to be published and the exact mechanism of action of this drug remains unknown at this time.

**Table 3 ijms-25-13321-t003:** Clinical trials of CRISPR–Cas-based gene therapies listed in the Australian New Zealand Clinical Trials Registry, Chinese Clinical Trial Register, and clinicaltrials.gov.

Clinical Trial ID	Disease/Condition	CRISPR–Cas System/Target	Modified Cells/Delivery	Reference
ACTRN12623001095651p	Atherosclerotic cardiovascular disease	CRISPR–Cas9 disrupting apolipoprotein (a)	Hepatocytes/in vivo by lipid nanoparticle	[60]
ACTRN12623000809639	Refractory dyslipidemias	CRISPR–Cas9 disrupting angiopoietin-like protein 3	Hepatocytes/in vivo by lipid nanoparticle	[64]
ChiCTR2300073795	Pyruvate kinase deficiency	CRISPR–Cas9 correction of pyruvate kinase	CD34+ cells/ex vivo by AAV6	[96]
ChiCTR2100041827	Osteoarthritis	CRISPR–Cas9 disruption of MMP-13	Chondrocytes/in vivo by exosomes	[66]
NCT03057912	HPV-related cervical intraepithelial neoplasia Ⅰ	CRISPR–Cas9 disruption of the E6/E7 viral component	Cervix squamous cells/in vivo by plasmids	Clinicaltrials.gov
NCT03164135	HIV-1 infection	CRISPR–Cas9 disruption of CCR5	CD34+ cellseEx vivo by electroporation	[73]
NCT04560790	Refractory herpetic viral keratitis	CRISPR–Cas9 disruption of the HSV-1 gnome	Stromal cells/in vivo by RNA injection	[82]
NCT05144386	HIV-1 infection	CRISPR–Cas9 targeting 3 sites of the viral genome	CD4+ cells/in vivo by AAV9	[97]
NCT05565248 NCT05210530	Diabetes mellitus 1	CRISPR–Cas9 disruption of B2M and TXNIP	Pancreatic endoderm cells/ex vivo	[68]
NCT06031727	Neovascular age-related macular degeneration	CRISPR–Cas13 disruption of the VEGF transcript	Subretinal injection of AAV	[95]
NCT06025032	Congenital hearing loss	CRISPR–Cas13	Intra cochlear injection of AAV	Clinicaltrials.gov
NCT05120830	Hereditary angioedema	CRISPR–Cas9 disruption of KLKB1	Hepatocytes/in vivo by lipid nanoparticle	[87]
NCT05805007	Retinitis pigmentosa	CRISPR–Cas9 correction of the RHO gene	Subretinal injection of AAV	[98]
NCT04601051	Cardiac amyloidosis	CRISPR–Cas9 disruption of the TTR gene	Hepatocytes/in vivo by lipid nano particle	[89]
NCT03872479	Leber congenital amaurosis	CRISPR–Cas9 disruption of the CEP290 gene	Subretinal injection of AAV5	[91]

## 5. CRISPR–Cas and Xenotransplantation

Xenotransplantation refers to any procedure that involves transplanting, implanting, or infusing live cells, tissues, or organs from an animal source into a human recipient. [99]. Xenotransplantation offers new hope for people on organ transplantation waiting lists.

The earliest record of xenotransplantation was a cornea transplantation trial that took place in 1838 [100]. Another xenotransplantation of a rabbit kidney to a human was done in 1905, and the patient died after 17 days due to graft rejection [101]. Between 1920–1990 many transplant studies were done using the kidneys, hearts, and livers of non-human primates [102], as well as other mammals, including rabbits, sheep, and pigs [103].

The modern concept of xenotransplantation focuses mainly on pigs as donors, pigs share physiological similarities with humans, reach sexual maturity within few months, and produce large litters. Moreover, they carry a lower risk of zoonotic disease transmission compared to nonhuman primates (NHPs) and can be bred under specific pathogen-free (SPF) conditions, which further minimizes infection risks [104]. Additionally, the use of primate organs is unsustainable due to ethical concerns and the endangered status of many primate species [105].

The Third WHO Global Consultation on Regulatory Requirements for Xenotransplantation Clinical Trials recommended that donor pigs should possess a minimum set of ‘essential’ genetic modifications [106]; those recommendations are based on three main principles:Knocking down genes that trigger immune responses in humans.Knocking in human transgenes to improve compatibility.Turning off pig endogenous retroviruses (PERVs).

To reduce acute rejection in xenotransplantation, the galactose-1,3-galactose epitope is knocked out by inactivating the *GGTA1* gene (alpha-1,3-galactosyltransferase). This modification is often combined with the knockout of the *B4GalNT2* gene (beta-1,4 N-acetylgalactosaminyltransferase 2) and the *CMAH* gene (cytidine monophosphate-N-acetylneuraminic acid hydroxylase) [107].

The Cas9 endonuclease and gRNA were used to create pigs lacking *GGTA1*, *GGTA1/CMAH*, or *GGTA1/CMAH/β4GalNT2* genes. Cells derived from modified pigs exhibited reduced human IgM and IgG binding [108].

To improve the compatibility of organs with the human immune system, by overcoming other obstacles like innate cellular xenograft rejection, coagulation dysregulation, and systemic inflammation human genes can be introduced into pigs, thus minimizing organ rejection.

The introduction of human transgenes such as *DAF*, *CD46*, and *CD59* has been effective in inhibiting complement activation. Additionally, coagulation inhibition can be achieved by integrating genes like human *CD39*, human thrombomodulin, and endothelial protein C receptor. Moreover, the inclusion of anti-inflammatory genes, such as *HO-1* and *A20*, alongside immunosuppressive molecules like *CTLA4Ig*, *CD47*, *PD-L1* and human Class I *MHC*, has demonstrated significant potential in overcoming immune barriers to xenotransplantation [109].

Recently, modified dCas technology is being utilized for the targeted knock-in of humanized genes while simultaneously knocking out previously mentioned genes [110]. Finally, CRISPR–Cas9 large-scale genome editing has been employed to inactivate the expression of porcine endogenous retroviruses (PERVs), which are found in multiple copies in the porcine genome [111].

The simplicity of gene editing with the CRISPR–Cas system has revitalized the field of xenotransplantation. This technology was utilized in the widely known recent heart xenotransplantation trial, where a 57-year-old patient named David Bennett, suffering from end-stage heart disease and ineligible for a conventional human heart transplant, received a pig heart with 10 genetic modifications. The patient survived for 60 days, with the transplanted heart beginning to fail on day 49 [112].

In contrast, pig kidney transplants were, until recently, limited to brain-dead patients. Montgomery et al. conducted a kidney xenotransplantation on a brain-dead patient, who survived for 32 days [113]. In 2024, a highly publicized case involved a kidney transplant to a living human, starting on March 21 and concluding on May 11, after the patient passed away without any evidence linking the cause of death to the transplanted organ [114]. This kidney had undergone 69 genetic modifications using the CRISPR–Cas9 system [111].

Xenotransplantation of other organs such as the liver, lungs, cornea, and islets remain in the preclinical trial phase and have not yet advanced to clinical studies [99].

## 6. Limitations, Challenges, and a New Perspective

### 6.1. Efficiency and Mosaicism

Despite recent improvements in CRISPR–Cas systems for genetic modification of all types of cells and tissues, the efficiency of this editing tool is still limited by various factors. These include the structure of the Cas protein, the sequence of the guide RNA, the availability of the target sequence (which is related to the modified cell type), and, finally, the delivery method of the CRISPR–Cas system, both in vivo and in vitro, which will be discussed later in this section [115,116].

Studies focused on modifying the Cas protein have yielded more capable variants with low off-target effects, such as SpCas9 (NGG) and Sniper-Cas9 [115]. Simultaneously, algorithms developed for guide RNA design and high-throughput screening tools have significantly reduced the time required to select the most efficient RNA sequences with minimal off-target effects [117].

Another limitation often associated with efficiency is mosaicism, especially in modified zygotes, tissues, and in vivo studies [118]. Mosaicism is characterized by an organism or specific tissue consisting of cells with different genetic compositions [119]. Although mosaicism can be linked to low editing efficiency, it is not limited to this factor. Mosaicism can result from various causes: the delay in Cas editing in proliferating cell types could lead to heterogeneity in the resulting daughter cells, while fast and persistent Cas protein activity can introduce new modifications in already edited cells. Additionally, the unpredictability of Cas-induced double-strand breaks (DSBs) and the cell’s repair mechanisms can result in a heterogeneous population of edited cells [120].

Mosaicism poses a significant challenge in germline editing, especially when the ultimate goal is achieving a modified organism that is homozygous for a targeted modification [121]. Furthermore, mosaicism limits the efficiency of all in vivo applications of CRISPR–Cas genome editing which is described in previous sections discussing various in vivo clinical applications of CRISPR–Cas systems.

On the other hand, ex vivo applications are minimally affected by either low efficiency or mosaicism. This is evident in the success of the CRISPR–Cas system in modifying hematopoietic progenitor cells followed by their infusion into patients [120].

### 6.2. Off-Target Effects

Despite the superior specificity and efficiency of CRISPR–Cas systems, off-target effects remain a significant challenge [122]. These effects occur when Cas proteins make unintended cuts in the genome, leading to unpredictable consequences. Off-target sites can be related to the sequence of the guide RNA, as Cas9 can tolerate up to three mismatches between the guide RNA and target DNA [123]. Such mismatches can result in DNA cleavage at regions highly similar to the intended target sequence. Additionally, sgRNA-independent off-target effects also exist, which can arise due to complex factors such as chromatin structure and sequence context [124].

More complex off-target effects, beyond point mutations and small insertions or deletions (indels), include structural variations (SVs) such as deletions, duplications, inversions, and translocations. These SVs can significantly disrupt the genome [125]. The leading cause is often improper repair of double-strand breaks (DSBs) caused by Cas9 at on or off-target sites [126]. These structural variations can interfere with normal gene function, potentially contributing to genomic instability. Improved prediction algorithms to generate specific guide RNA, as well as the discovery of new Cas types and variants can aid in evading off-target effects and improving efficiency [127].

Cas proteins efficient induction of double-strand breaks is almost bound to exhibit some levels of non-target effects or lead to random correction of the breaks. That is why implementing alternative approaches is an active field of investigation.

Base editors, consisting of a Cas enzyme for targeted DNA binding and a single-stranded DNA-modifying enzyme for nucleotide alteration, and prime editing, which combines an engineered reverse transcriptase fused to Cas9 nickase with a prime-editing guide RNA (pegRNA) [128], are the most promising alternatives. Both have shown encouraging results, leading to their entry into clinical trials very recently.

Currently, there is only one ongoing preclinical study investigating the efficiency of a prime editing-based therapy for chronic granulomatous disease (CGD), an inherited primary immunodeficiency characterized by recurrent infections due to a lack of nicotinamide adenine dinucleotide phosphate (NADPH) oxidase activity in phagocytic myeloid cells such as neutrophils. The most common mutation leading to autosomal recessive CGD is located in the NCF1 gene, which encodes the p47phox protein, a subunit of the NADPH oxidase complex. The most common mutation in this gene is a 2-nucleotide GT deletion (delGT) in exon 2 of NCF1. Preclinical results of PM359 showed restoration of p47phox protein expression and NADPH oxidase activity in prime-edited patient CD34+ cells [129]. As a result, a clinical trial (NCT06559176) was approved and initiated in November 2024.

Base editing is more widely used in genome editing studies, with multiple therapies already entering the clinical trial stage. Beam Therapeutics is already testing a base editing ex vivo therapy that aims to correct the single nucleotide variation in hematopoietic stem cells leading to SCD [130].

Beam-301, developed by the same company, is a base editing genetic therapy aiming to treat glycogen storage disease type Ia, which is caused by a single nucleotide change giving rise to the p.R83C variant of glucose-6-phosphatase-α (G6PC), leading to the inability to store glycogen in the liver and hence not being able to maintain blood sugar levels [131].

Another drug from Beam Therapeutics, BEAM-302, entered the clinical trial stage in 2024 and is set to correct the mutation in the alpha-1 antitrypsin gene, leading to the expression of the p.Glu366Lys form, which misfolds and aggregates inside liver cells, causing liver damage and alpha-1 antitrypsin deficiency (AATD) [132]. Both drugs, BEAM-301 and BEAM-302, use the injection of liver-targeting lipid nanoparticles to deliver the base editor to the targeted cells [131,132].

Beam Therapeutics is not the only player in base editing therapy, as VERVE Therapeutics, another US-based company, is working on two therapies for the treatment of heterozygous and homozygous familial hypercholesterolemia. The mechanism of both drugs uses base editing for modifying and inactivating the targeted proprotein convertase subtilisin/kexin type 9 (*PCSK9*) gene, which plays a key role in lipid metabolism, leading to a durable reduction in LDL blood levels. Their editing system is delivered in vivo to the targeted cells by lipid nanoparticles [133,134].

### 6.3. Delivery Challenges

Delivering CRISPR–Cas9 components to target cells presents significant challenges, particularly in ensuring safe, efficient, and specific delivery. Viral vectors, such as (AAVs), are commonly used for in vivo delivery due to their ability to transduce cells efficiently and provide long-lasting gene expression [135]. However, viral vectors come with their own limitations. For example, a recent clinical trial (NCT05514249) for duchenne muscular dystrophy resulted in the death of a participant after receiving CRD-TMH-001, a CRISPR–Cas9 therapy delivered by AAV6. The patient developed an acute immune response, likely related to the high dose of rAAV used in the trial. This case highlights the potential risks associated with viral vector delivery [136]

Additionally, the upper limit of AAV packaging is 4.7 kb, which significantly restricts the capacity of the CRISPR/Cas system to load genes [137]. This limits the size of the Cas protein used, such as SpCas9 derived from Streptococcus pyogenes, which has a molecular mass of 4.1 kb, or necessitates the use of smaller Cas9 proteins with lower efficiency. Although dual AAV systems are being developed to overcome this limitation, higher doses of AAV are more likely to induce adverse effects in clinical trials, as discussed earlier [138].

Non-viral methods, such as lipid nanoparticles (LNPs) or electroporation, are more commonly used in in vitro settings, particularly for editing immune cells or stem cells before reinfusion [139]. Despite having low immunogenicity, these delivery methods face challenges, such as reduced transfection efficiency in certain cell types, and the low specificity when used in precise tissue-specific targeting [140]. Another limitation of non-viral delivery methods is their short-term stability, as CRISPR components delivered via nanoparticles or electroporation may degrade rapidly in the bloodstream, reducing the overall efficacy of therapeutic interventions [141].

### 6.4. Immunogenicity of Cas Protiens

Immunogenicity of CRISPR–Cas-based therapies is not conclusive to the viral components of the delivery vector as discussed above. Wang et al. detected IgG1, IgG2a, and IgG2b antibodies against Cas9 in mouse models with AAV delivery of Cas9 for hepatocyte-specific editing [142]. Furthermore, a screening study detected antibodies against SpCas9 and SaCas9 in the serum of human donors [143]. Despite no safety problems observed in ongoing clinical trials, immunizing mice against the Cas9 protein resulted in hepatocyte apoptosis and failure of genome editing after the injection of an AAV vector expressing the same Cas9 protein [144]. Taken together, the immunogenicity of Cas9 can lead to low editing efficiency and potentially serious adverse events in the case of in vivo gene editing using the CRISPR–Cas system. This has driven the development of various strategies to evade the immune response of the host. Currently implemented approaches including, masking immunogenic Cas9 epitopes, utilizing Cas9 orthologs from non-pathogenic bacteria and inducing immune tolerance [145].

## 7. Conclusions

CRISPR/Cas technology has marked a pivotal shift in genetic medicine, enabling precise, targeted treatments for previously intractable genetic disorders. The rapid expansion of the CRISPR/Cas application range in clinical trials reflects the great potential of this technology in gene therapy, particularly for monogenic diseases. However, due to its relatively recent use, the long-term effects of CRISPR/Cas are still poorly understood. Described limitations of the system can lead to delayed adverse effects, especially in cases of the unintentional editing of germ cells, which may manifest in later generations and necessitate some of the longest follow-up trials among all developed drugs. Advances in trial design, gene editing precision, and other biomedical fields, especially immunology, will enhance the safety and efficacy of this technology. As research progresses across biomedical sciences, the range of possible applications for CRISPR–Cas-based therapies will likely prove to be limitless.

## Figures and Tables

**Figure 1 ijms-25-13321-f001:**
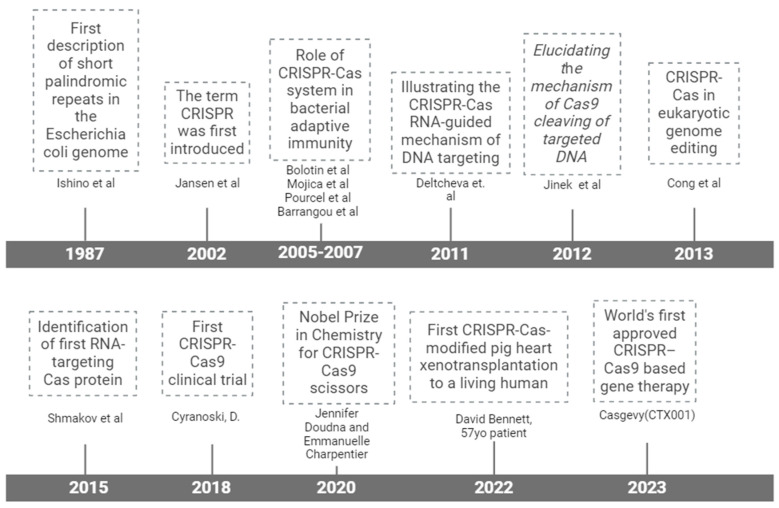
Milestones in the history of CRISPR–Cas technology from its discovery in 1987 until the first FDA-approved gene therapy using CRISPR–Cas9 for genome modification: Ishino [21], Jansen [22], Bolotin [23], Mojica [24], Pourcel [25], Barranougou [26], Deltcheva [27], Jinek [1], Cong [28], Shmakov [29], Cyranoski [30].

**Figure 2 ijms-25-13321-f002:**
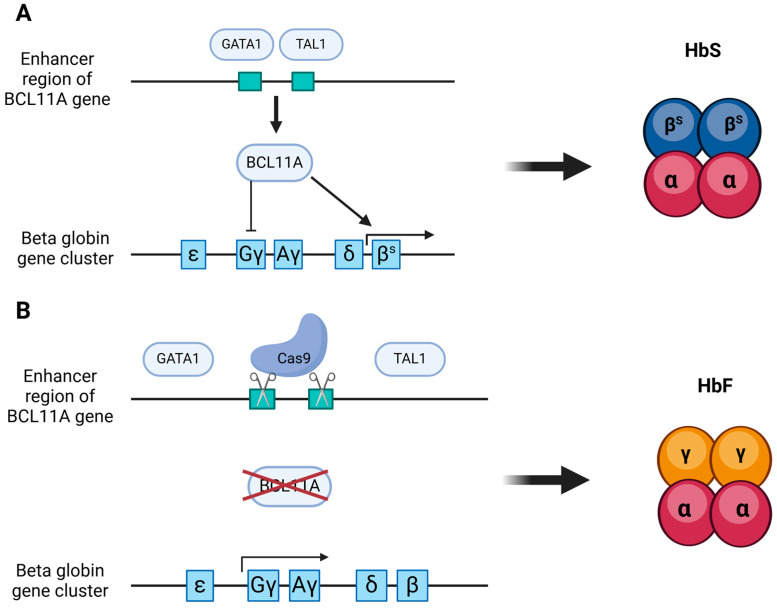
CTX001 mechanism of action. (**A**) BCL11A promotes βs-globin expression while suppressing γ-globin expression, resulting in HbS production and the formation of crescent-shaped cells. (**B**) Knockdown of BCL11A, achieved by introducing a cut at the binding region of transcription factors GATA1/TAL1, enables γ-globin gene expression, leading to HbF production and the development of healthy erythrocytes.

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
