# Peer review of "Expanding Horizons of CRISPR/Cas Technology: Clinical Advancements, Therapeutic Applications, and Challenges in Gene Therapy"

_ijms, 2024, doi:10.3390/ijms252413321_

Round 1

Reviewer 1 Report

Comments and Suggestions for Authors

This article presents a comprehensive review of CRISPR/Cas technology, including its therapeutic applications. The discussion is timely and relevant, particularly in highlighting recent clinical trials and emerging applications in gene therapy. 

At the same time, I would suggest a few enhancements to enhance it:

Part 1

Consider discussing the clinical applications of CRISPR systems that do not induce double-strand breaks, particularly base editing, which is currently being used in clinical trials. The role of base editing should be appropriately highlighted in the clinical applications of CRISPR.

Part 4

1) Section 4.1 provides a concise summary of various aspects of CRISPR applications in cancer, helping readers quickly grasp the key points. It is suggested that similar summaries be included in Sections 4.2 and 4.3 for consistency and ease of understanding.

2) The CCR5 knockout strategy in Section 4.4 cannot be categorized into either of the two groups described in line 211. Please clarify or revise the categorization to avoid confusion.

Highlighting Delivery Challenges:

   The section on delivery challenges could highlight the limitations imposed by the packaging capacity of delivery vectors, such as adeno-associated viruses (AAVs). Current AAV vectors struggle to accommodate large Cas proteins and complex editing components, which constrains their applicability in advanced therapies. Discussing alternative delivery strategies would add depth to this section.

Author Response

For review article

Response to Reviewer 1 Comments

1. Summary

Thank you very much for taking the time to review this manuscript.

  1. Point-by-point response to Comments and Suggestions for Authors

Comments 1:

This article presents a comprehensive review of CRISPR/Cas technology, including its therapeutic applications. The discussion is timely and relevant, particularly in highlighting recent clinical trials and emerging applications in gene therapy.

At the same time, I would suggest a few enhancements to enhance it:

Part 1

Consider discussing the clinical applications of CRISPR systems that do not induce double-strand breaks, particularly base editing, which is currently being used in clinical trials. The role of base editing should be appropriately highlighted in the clinical applications of CRISPR.

Part 4

1) Section 4.1 provides a concise summary of various aspects of CRISPR applications in cancer, helping readers quickly grasp the key points. It is suggested that similar summaries be included in Sections 4.2 and 4.3 for consistency and ease of understanding.

2) The CCR5 knockout strategy in Section 4.4 cannot be categorized into either of the two groups described in line 211. Please clarify or revise the categorization to avoid confusion.

Highlighting Delivery Challenges:

   The section on delivery challenges could highlight the limitations imposed by the packaging capacity of delivery vectors, such as adeno-associated viruses (AAVs). Current AAV vectors struggle to accommodate large Cas proteins and complex editing components, which constrains their applicability in advanced therapies. Discussing alternative delivery strategies would add depth to this section.

Response 1:

Thank you very much for your valuable and encouraging review. Your comments helped us revisit some missed points in the original paragraph and enhanced the scientific accuracy of the text.

Part 1:

We added a paragraph to subsection 6.2, where we discuss off-target effects and suggest both prime editing and base editing as promising alternatives to double-strand breaks induced by the Cas9 protein. Additionally, we described ongoing clinical trials using prime editing and base editing for genome modification, summarizing their mechanisms and the targeted genes in a manner consistent with the systems we referenced throughout the article. We preferred to include this paragraph in subsection 6.2, as our article is categorized by types of treated diseases. However, we are open to suggestions if you believe this paragraph would fit better in another part of the review.

Part 4:

We added small summaries to subsections 4.1 and 4.2. lines (118-121) and (176-181) respectively

We clarified the strategy used in CCR5 knockout, ensuring it stands out as distinct from the strategy used by Excision BioTherapeutics.  Subsection 4.4 Lines (235-242)

Highlighting Delivery Challenges:

We added a brief paragraph discussing the size limitations imposed by the use of AAV delivery methods. In the original draft, non-viral delivery methods are also discussed as an alternative, with a clear explanation of their advantages and disadvantages. Subsection 6.3 (Lines 485-491)

Reviewer 2 Report

Comments and Suggestions for Authors

XX et al.'s review article, entitled “Expanding Horizons of CRISPR/Cas Technology: Clinical Advancements, Therapeutic Applications, and Challenges in Gene Therapy,” is comprehensive and covers recent advancements in clinical trials and preclinical studies. While the reviewer recommends its publication in YY, the section addressing challenges could be strengthened. The current version discusses two: the delivery and off-target effect, but overlooks several important aspects, such as limitations in editing efficiency, potential mosaicism, long-term effects, and the immunogenicity of Cas proteins. Furthermore, the application of gene editing in autologous cell-based therapy should be included to provide a more complete perspective.

Author Response

For review article

Response to Reviewer 2 Comments

1. Summary

Thank you very much for taking the time to review this manuscript.

2. Point-by-point response to Comments and Suggestions for Authors

Comments 1:

[XX et al.'s review article, entitled “Expanding Horizons of CRISPR/Cas Technology: Clinical Advancements, Therapeutic Applications, and Challenges in Gene Therapy,” is comprehensive and covers recent advancements in clinical trials and preclinical studies. While the reviewer recommends its publication in YY, the section addressing challenges could be strengthened. The current version discusses two: the delivery and off-target effect, but overlooks several important aspects, such as limitations in editing efficiency, potential mosaicism, long-term effects, and the immunogenicity of Cas proteins. Furthermore, the application of gene editing in autologous cell-based therapy should be included to provide a more complete perspective.]

Response 1:

Thank you very much for taking the time to read this review thoroughly and providing valuable comments, which helped in revisiting some points that were overlooked during the writing of the original draft.

We have added a new subsection (6.1) discussing editing efficiency and potential mosaicism in CRISPR-Cas applications. Furthermore, we added subsection 6.4, which addresses the immunogenicity of Cas proteins as a challenge in clinical implementations.

The application of gene editing in autologous cell-based therapy is described in various sections, with hematological disorders being the field of preference. The idea of autologous cell modification can be inferred from the mechanism of action of various therapies. However, we have added a summary to the section on hematological disorders, emphasizing the importance of the successful modification of autologous hematopoietic stem cells. Finally, the long-term effects of CRISPR-Cas genetic modification are still poorly understood, especially in patients after certified therapy or clinical trials. However, we fully agree on the importance of addressing this risk, which remains concealed within the novelty of the system. We have emphasized it in the conclusion as a consequence of the combined limitations of CRISPR-Cas technology in its current applied form. Please let us know if any further adjustments are needed or if we missed any aspects of the described sections.
